# A retrospective evaluation of access equity in virtual care during the COVID-19 pandemic: A 2-year review and comparison of visits in Ontario, Canada

Shawn Mondoux[1,2,3]*, Frank Battaglia[1], Anastasia Gayowsky[4], Natasha Clayton[3,5], Caillin Langmann[1,3], Paul Miller[1,5], Alim Pardhan[1,5], Julie Matthews[5], Alexander Drossos[6], Keerat Grewal[7,8,4]

1 Department of Medicine, Division of Emergency Medicine, McMaster University, Hamilton, Ontario, Canada, 2 Institute of Health Policy, Management and Evaluation, Dalla Lana School of Public Health, University of Toronto, Toronto, Ontario, Canada, 3 St. Joseph's Healthcare Hamilton, Hamilton, Ontario, Canada, 4 ICES, Toronto, Ontario, Canada, 5 Hamilton Health Sciences, Hamilton, Ontario, Canada, 6 Department of Psychiatry and Behavioral Neurosciences, McMaster University, Hamilton, Ontario, Canada, 7 Schwartz/Reisman Emergency Medicine Institute, Sinai Health, Toronto, Ontario, Canada, 8 Division of Emergency Medicine, Department of Medicine, University of Toronto, Toronto, Ontario, Canada

* shawn.mondoux@medportal.ca

## Abstract

### Background

Access equity has been raised as a fundamental concern with virtual care, both as it was used during the SARS-CoV-2 pandemic as well as its future applications within health systems. These concerns have not yet been substantiated with quantifiable data. We conducted a comparison of healthcare utilization and access across all dimensions of the Ontario Marginalization Index between virtual care and in-person care in the province of Ontario.

### Methods

We conducted a retrospective observational study using ICES databases in the Province of Ontario between March 14, 2020, and March 13,2022. We identified all virtual and in-person visits using billing codes. All visits were linked to their individual postal dissemination area for which there was census data from the Ontario Marginalization Index. Dissemination areas were divided, according to their categorization within each marginalization dimension, and visit rates were calculated for both populations.

### Results

A total of 93,363,194 visits were included as part of the final analysis. Significant differences in virtual healthcare utilization were noted between the most and least marginalized populations within each dimension. This effect was not observed by visits

**Data availability statement:** The dataset from this study is held securely in coded form at ICES. While data sharing agreements prohibit ICES from making the dataset publicly available, access may be granted to those who meet pre-specified criteria for confidential access, available at www.ices.on.ca/DAS. The full dataset creation plan and underlying analytic code are available from the authors upon request or may be requested from ICESMcMasterAdmin@ices.on.ca, understanding that the computer programs may rely upon coding templates or macros that are unique to ICES and are therefore either inaccessible or may require modification.

**Funding:** Declaration of Interest: SM was the recipient of the Juravinski Research Institute grant which paid for the costs of administrative data use. SM was also the recipient of the Graham Farquharson Knowledge Translational Fellowship, administered by PSI, which created protected time for research. NC was a paid research coordinator as part of this project. SM, FB, CL, PM, AP, AD and KG are all clinicians providing clinical care using both virtual and in-person modalities throughout the pandemic. AG is a paid employee of the Institute of Clinical Evaluative Sciences (ICES).

**Competing interests:** The authors have declared that no competing interests exist.

for in-person care. The only exception was that racialized, and newcomer populations had higher virtual care utilization among the most marginalized.

## Interpretation

This data is the first that uses a large retrospective dataset and seems to confirm concerns for access inequity among the most marginalized populations. These differences much be part of policy considerations for the future of virtual care use.

---

## Introduction

The landscape of healthcare delivery is undergoing a transformative shift, driven by advancements in technology and the evolving needs of patients and providers [1]. In particular, the adoption of virtual care services has witnessed unprecedented growth, offering a promising avenue to enhance healthcare accessibility, convenience, and efficiency. As a vast and geographically diverse country, Canada faces unique challenges in ensuring equitable access to healthcare services, exacerbated by factors such as geographic isolation, socio-economic disparities, and varying levels of digital infrastructure. Within the scope of this paper, virtual care is defined as care offered through telephone or internet-enable video conferencing tools. The COVID-19 pandemic accelerated the adoption of virtual health care services, because of their potential to bridge gaps in access to healthcare, reduce healthcare system strain, and safeguard vulnerable populations during the pandemic [1,2]. However, understanding the benefits of virtual care for all Canadians remains a complex and evolving challenge.

Yet studies exploring the equity in virtual care access have been limited. Case studies, narratives and qualitative experiences have been published which describe the experience or perspectives of specific groups as they relate to virtual care access [3–6]. National guidance documents were created by experts with the hope that specific strategies may be deployed to reduce access inequity [7]. Despite these publications, there is currently limited quantitative data assessing virtual care access equity in Canada or internationally.

Studies from our co-authors have included a thorough examination of the current state of virtual care adoption as well as short term patient outcomes [8,9]. In the current study, we aimed to explore factors influencing the accessibility of virtual care and disparities faced by various population groups. Specifically, we examined the relationship between the Ontario Marginalization Index (ONMARG) [10] (its summary score as well as access to material resources, age and labour force, racialization and newcomer status, and housing) and patient access to virtual care services. By examining the variability in virtual care access, this research aims to provide valuable insights that may serve as the foundation for evidence-based recommendations to healthcare policymakers, practitioners, and stakeholders in Canada.

## Methods

### Setting and population

We conducted a retrospective observational study using administrative health databases (held at ICES, formerly the Institute for Clinical Evaluative Sciences) in Ontario, Canada's most populous province with 15 million inhabitants. Due to our interest in examining the SARS-CoV2 pandemic's effect on virtual care in ambulatory care visits and patient outcomes, we compared virtual care visits and in-person care visits. The analytical pandemic timeframe was limited from March 14, 2020 (the day of pandemic declaration in Ontario) to March 13, 2022. A washout period of 30 days was included for each visit which ensured downstream outcomes for most categories were not included as subsequent index visits. We excluded index visits for outpatient surgical procedures, chemotherapy, radiation or COVID vaccinations using billing data. Virtual care visits were distinguished from in-person visits using specialized virtual care billing codes which existed both in the early pandemic (through historical virtual billing codes) and the intra-pandemic environments. To identify in-person visits, a complete review of the schedule of benefits was conducted to include only in-person physician encounter billing codes. This approach was used due to a very high proportion of "undefined" location codes within ICES billing data. All temporal changes to billing codes made during this period by the Ministry of Health (MoH) were included in the data analysis. Patients without a valid Ontario health card were excluded from the analysis. The analysis was done on a per-visit basis meaning that a patient may be included on several instances within the data set. This was done as patients may appear in both virtual and in-person populations within this time frame making their attribution unclear.

The ONMARG summary score, as well as its component dimensions were used for each patient visit. ONMARG is an area-based index that seeks to demonstrate marginalization between geographic areas and understand inequalities in measures of health and social well-being [10] and is already collected for all patients within our provincial administrative database. The four component dimensions of the ONMARG score are:

| Material Resources | Closely connected to poverty and refers to the inability for individuals and communities to access and attain basic material needs relating to housing, food, clothing, and education. |
|---|---|
| Age and Labour Force | Relates to area-level concentrations of people who don't have income from employment, including older adults, children, and/or those unable to work due to disability. |
| Racialized and Newcomer Populations | Measures the proportion of newcomers, and/or non-white, non-Indigenous populations. These communities may experience high levels of racialization and xenophobia. |
| Households and Dwellings | Relates to family and neighbourhood stability and cohesiveness and is based on measures of the types and density of residential accommodations and family structure characteristics. |

ONMARG summary scores, as well as each of the dimension scores, are available for each patient. Each of these scores range from 1 to 5–1 corresponding to the least marginalization in the dimension and 5 corresponding to the greatest.

### Data sources

The Ontario Health Insurance Plan (OHIP) database contains physician billing data for medically necessary care. This database also contains information on type of visit (i.e., inpatient, surgical, outpatient). The Same Day Surgery (SDS) database was used to capture outpatient surgery for cohort exclusion purposes. The Cancer Activity Level Reporting (ALR) database was used to capture cancer therapy visits for cohort exclusion purposes. The ONMARG dataset was used to ascribe Ontario Marginalization Index dimensions and summary scores to each patient. The Registered Persons Database (RPDB) contains validated mortality information for all Ontario residents, including out-of-hospital deaths. Datasets were linked using unique encoded identifiers and analyzed at ICES. Ontario has universal health care coverage for medically necessary care; therefore, these databases contain most of the healthcare utilization data in the province. All data were extracted for analysis in March 2023.

## Outcome measures and analyses

Dissemination areas are the smallest standardized geographic units for which census data are made available and are the unit of analysis used in this study of which there are 111 in the province of Ontario [11]. Within the ONMARG dataset, each dissemination area has its own ONMARG summary score and dimension scores. All dissemination areas within the province of Ontario were divided into one of the five categories of marginalization for each dimension of the ONMARG score. Within each of these dissemination areas, rates of visits were calculated for both in-person and virtual care. The primary outcome was the mean visit rate per 1000 residents within the dissemination area. The percentile-based bootstrap method was used to create N = 1000 samples at the dissemination area level, from which we calculated 95% confidence intervals per quintile by taking the 2.5% and 97.5% percentile points from the N = 1000 bootstrap samples.

## Ethics and approval

ICES's collection and use of data is authorized under Section 45 of Ontario's Personal Health Information Protection Act (PHIPA) as a prescribed entity, which is exempt from review by a Research Ethics Board [12,13]. The use of the data in this study is authorized under section 45 and approved by ICES's Privacy and Legal Office. All retrospective data held at ICES is fully anonymized to the authors and use of ICES data does not require informed consent from individual patients.

## Results

### Study population

After exclusions, there were 47,092,297 visits in the virtual care visits and 46,270,897 visits in the in-person care visits who were eligible for inclusion (for a total of 93,363,194 visits). A comparison for population demographics is found in Table 1. Table 2 outlines the exclusion criteria applied as well as the quantitative results of these exclusions. Patient visits required a valid health insurance number and billing data on the visit which had a specialized date.

### Age and labour force factor

Within this dimension of the ONMARG index, marginalization refers to the number of individuals without employment-based income within a dissemination area and relates to the impacts of disability and dependence [14]. With respect to in-person visit rates, we found a linear relationship between age and labour force marginalization and in-person visits, with patients who are most marginalized having the highest rate of in-person healthcare visits while those who were least marginalized having a lower rate of in-person healthcare visits.

This trend was reversed when examining virtual care use, with the least marginalized patients within this category having the highest virtual care visit rates. The only exception in this finding was that patients in category 5 (highest marginalization) used virtual care at a greater rate than those in category 4.

### Households and dwellings factor

Within this dimension of the ONMARG index, marginalization refers to areas of low family and neighborhood stability and cohesiveness. This dimension is related to the types and densities of residential accommodations as well as some characteristics of family structure within an area [14]. With respect to in-person visits, similar use frequencies are seen across the 5 categories, with all levels of marginalization having similar rates of in-person visits.

Within virtual care, there is significant variability with those that are least marginalized (in the most cohesive neighborhoods) using virtual care the most. Categories 2 and 5 (among the least and the most marginalized) have utilization in similar rates while those in categories 3 and 4 have the lowest rates of use.

**Table 1. Comparison of patient demographics between virtual care and in-person care cohorts.**

| Variable | Value | Virtual care visits N = 47,092,297 | In-person care visits N = 46,270,877 | Standardized Difference | P-value |
|---|---|---|---|---|---|
| Age category | 0-6 | 1,941,804 (4.1%) | 3,636,791 (7.9%) | **0.158** | <.0001 |
| | 7-12 | 1,304,694 (2.8%) | 1,533,611 (3.3%) | 0.032 | |
| | 13-17 | 1,462,018 (3.1%) | 1,573,726 (3.4%) | 0.017 | |
| | 18-34 | 8,660,774 (18.4%) | 8,195,346 (17.7%) | 0.018 | |
| | 35-44 | 6,180,077 (13.1%) | 5,441,489 (11.8%) | 0.041 | |
| | 45-54 | 6,727,449 (14.3%) | 5,756,500 (12.4%) | 0.054 | |
| | 55-64 | 8,028,981 (17.0%) | 7,237,656 (15.6%) | 0.038 | |
| | 65-74 | 6,953,829 (14.8%) | 6,717,397 (14.5%) | 0.007 | |
| | 75-84 | 4,178,024 (8.9%) | 4,324,428 (9.3%) | 0.016 | |
| | 85+ | 1,654,647 (3.5%) | 1,853,933 (4.0%) | 0.026 | |
| Sex | F | 27,316,159 (58.0%) | 26,029,074 (56.3%) | 0.035 | <.0001 |
| | M | 19,776,138 (42.0%) | 20,241,803 (43.7%) | 0.035 | |
| LHIN | 1 | 2,034,019 (4.3%) | 2,381,410 (5.1%) | 0.039 | <.0001 |
| | 2 | 2,622,828 (5.6%) | 3,202,449 (6.9%) | 0.056 | |
| | 3 | 2,137,267 (4.5%) | 2,325,661 (5.0%) | 0.023 | |
| | 4 | 4,803,791 (10.2%) | 5,368,293 (11.6%) | 0.045 | |
| | 5 | 4,110,502 (8.7%) | 3,447,492 (7.5%) | 0.047 | |
| | 6 | 4,875,182 (10.4%) | 4,014,076 (8.7%) | 0.057 | |
| | 7 | 4,587,029 (9.7%) | 3,909,086 (8.4%) | 0.045 | |
| | 8 | 7,105,197 (15.1%) | 6,541,335 (14.1%) | 0.027 | |
| | 9 | 5,852,216 (12.4%) | 5,249,345 (11.3%) | 0.033 | |
| | 10 | 1,326,543 (2.8%) | 1,637,557 (3.5%) | 0.041 | |
| | 11 | 4,265,569 (9.1%) | 4,019,468 (8.7%) | 0.013 | |
| | 12 | 1,432,389 (3.0%) | 1,566,351 (3.4%) | 0.019 | |
| | 13 | 1,375,842 (2.9%) | 1,874,052 (4.1%) | 0.062 | |
| | 14 | 563,923 (1.2%) | 734,302 (1.6%) | 0.033 | |
| Rural status | Urban | 43,461,017 (92.3%) | 41,839,636 (90.4%) | 0.066 | <.0001 |
| | Rural | 3,523,911 (7.5%) | 4,312,068 (9.3%) | 0.066 | |
| | Missing | 107,369 (0.2%) | 119,173 (0.3%) | 0.006 | |
| Income quintile | 1 (lowest) | 8,947,793 (19.0%) | 9,389,913 (20.3%) | 0.033 | <.0001 |
| | 2 | 9,260,862 (19.7%) | 9,190,581 (19.9%) | 0.005 | |
| | 3 | 9,596,326 (20.4%) | 9,344,766 (20.2%) | 0.005 | |
| | 4 | 9,603,901 (20.4%) | 9,244,316 (20.0%) | 0.010 | |
| | 5 (highest) | 9,562,322 (20.3%) | 8,965,024 (19.4%) | 0.023 | |
| | Missing | 121,093 (0.3%) | 136,277 (0.3%) | 0.007 | |
| Dependency Quintile | 1 (lowest) | 13,428,697 (28.5%) | 12,199,592 (26.4%) | 0.048 | <.0001 |
| | 2 | 9,553,055 (20.3%) | 9,115,790 (19.7%) | 0.015 | |
| | 3 | 7,963,239 (16.9%) | 7,928,223 (17.1%) | 0.006 | |
| | 4 | 7,543,068 (16.0%) | 7,777,546 (16.8%) | 0.021 | |
| | 5 (highest) | 8,309,070 (17.6%) | 8,838,326 (19.1%) | 0.038 | |
| | Missing | 295,168 (0.6%) | 411,400 (0.9%) | 0.030 | |
| Deprivation Quintile | 1 (lowest) | 11,067,378 (23.5%) | 10,189,296 (22.0%) | 0.035 | <.0001 |
| | 2 | 9,942,284 (21.1%) | 9,534,539 (20.6%) | 0.012 | |
| | 3 | 8,943,468 (19.0%) | 8,698,461 (18.8%) | 0.005 | |
| | 4 | 8,439,720 (17.9%) | 8,502,804 (18.4%) | 0.012 | |
| | 5 (highest) | 8,404,279 (17.8%) | 8,934,377 (19.3%) | 0.038 | |
| | Missing | 295,168 (0.6%) | 411,400 (0.9%) | 0.030 | |

*(Continued)*

**Table 1.** (Continued)

| Variable | Value | Virtual care visits N = 47,092,297 | In-person care visits N = 46,270,877 | Standardized Difference | P-value |
|---|---|---|---|---|---|
| Ethnic Diversity Quintile | 1 (lowest) | 6,124,793 (13.0%) | 7,064,983 (15.3%) | 0.065 | <.0001 |
| | 2 | 6,944,635 (14.7%) | 7,528,699 (16.3%) | 0.042 | |
| | 3 | 8,282,606 (17.6%) | 8,182,896 (17.7%) | 0.003 | |
| | 4 | 10,419,490 (22.1%) | 9,650,429 (20.9%) | 0.031 | |
| | 5 (highest) | 15,025,605 (31.9%) | 13,432,470 (29.0%) | 0.063 | |
| | Missing | 295,168 (0.6%) | 411,400 (0.9%) | 0.030 | |
| Instability Quintile | 1 (lowest) | 10,823,300 (23.0%) | 9,840,737 (21.3%) | 0.041 | <.0001 |
| | 2 | 8,595,220 (18.3%) | 8,499,072 (18.4%) | 0.003 | |
| | 3 | 8,236,263 (17.5%) | 8,366,837 (18.1%) | 0.015 | |
| | 4 | 8,102,017 (17.2%) | 8,357,246 (18.1%) | 0.022 | |
| | 5 (highest) | 11,040,329 (23.4%) | 10,795,585 (23.3%) | 0.003 | |
| | Missing | 295,168 (0.6%) | 411,400 (0.9%) | 0.030 | |
| ONMARG summary score quintile | 1 (lowest) | 9,096,462 (19.3%) | 8,641,171 (18.7%) | 0.016 | <.0001 |
| | 2 | 9,410,432 (20.0%) | 8,975,642 (19.4%) | 0.015 | |
| | 3 | 9,381,610 (19.9%) | 9,179,327 (19.8%) | 0.002 | |
| | 4 | 9,263,307 (19.7%) | 9,239,988 (20.0%) | 0.007 | |
| | 5 (highest) | 9,645,318 (20.5%) | 9,823,349 (21.2%) | 0.018 | |
| | Missing | 295,168 (0.6%) | 411,400 (0.9%) | 0.030 | |
| Considered frail (ACG macro) | | 1,344,654 (2.9%) | 1,576,723 (3.4%) | 0.032 | <.0001 |
| Long-term care resident | | 57,035 (0.1%) | 102,955 (0.2%) | 0.024 | <.0001 |
| Total ADG score category | 0 | 2,056,271 (4.4%) | 2,169,788 (4.7%) | 0.016 | <.0001 |
| | 1-3 | 17,872,236 (38.0%) | 17,848,399 (38.6%) | 0.013 | |
| | 4-6 | 15,988,318 (34.0%) | 15,422,815 (33.3%) | 0.013 | |
| | 7-10 | 8,854,371 (18.8%) | 8,463,178 (18.3%) | 0.013 | |
| | 11+ | 2,321,101 (4.9%) | 2,366,697 (5.1%) | 0.009 | |

**Table 2. Exclusion criteria applied to patient populations.**

| Inclusion/Exclusion | Included | Excluded |
|---|---|---|
| **Virtual care visits pre-exclusions** | **91,801,929** | |
| Unique on IKN and record date | 76,051,175 | 15,750,754 |
| 30-day washout period | 47,203,634 | 28,847,541 |
| **In-person care visits pre-exclusions** | **81,326,940** | |
| Unique on IKN and record date | 78,017,303 | 3,309,637 |
| 30-day washout period | 46,421,792 | 31,595,511 |
| **Combined populations 1 & 2** | **93,625,426** | |
| Non-Ontario postal code on service date, or death date occurring prior to service date | 93,363,194 | 262,232 |
| **Full cohort after above exclusions** | **94,767,868** | |
| Virtual care visits | 47,092,297 | 111,337 |
| In-person care visits | 46,270,897 | 150,895 |
| **Total number of patients included** | **93,363,194** | |

## Material resources factor

Within this dimension of the ONMARG index, marginalization refers to elements closely related to poverty and attempts to gauge the inability for individuals and communities to access and attain basic material needs relating to housing, food, clothing, and education [14]. With respect to in-person visits, there was similar rates of use across all 5 categories of marginalization with overlap of many of the confidence intervals in each category, indicating similar rates of in-person visits across patients in all marginalization quintiles. Despite this, the most marginalized (categories 4 and 5) had higher rates of in-person visits.

Virtual care utilization rates were reversed, with those who were least materially marginalized using virtual care at a higher rate than more marginalized populations. Within this dimension in virtual care, there was an inverse linear relationship with regards to virtual care rates, with patients in category 5 (most marginalized) having the lowest rates of virtual care utilization.

## Racialized and newcomer populations factor

Within this dimension of the ONMARG index, marginalization refers to measures of the proportion of newcomers, and/or non-white, non-Indigenous populations. These are communities that may experience large amounts of racialization and xenophobia [14]. With respect to in-person care, there were higher rates of in-person visits within the least marginalized and racialized patients with minimal. The lowest rates of in-person visits are among the most racialized populations.

This trend was reversed for virtual visit populations. Within virtual care, the most marginalized (racialized) populations used virtual care at the highest rate with a linear relationship to low use in the least marginalized populations. This dimension demonstrates the most significant difference in visits rates between marginalization categories for virtual care among all dimensions.

## ONMARG summary score

Combining all the dimensions above into a summary score, we found that for in-person visits the most marginalized populations have the highest rates of in-person visits. Patients in categories 3 and 4 had the lowest rates of in-person visits with higher rates of visits among patients in categories 1 and 2.

With regards to virtual care visits, the opposite trend was seen, with more marginalized patients having the lowest rates of virtual care use with increasing use as patients are less marginalized. Category 5 (the most marginalized patients) used virtual care at rates that fall into the middle of all categories.

A visual depiction of all results can be seen in Fig 1.

## Interpretation

In this study, we found that during the COVID-19 pandemic there was less variability in the rates of in-person care across the different components of marginalization compared to rates of virtual care visits. There were, however, increased rates of in-person care noted among patients with less employment-based income (with more disability or dependence) as well as those patient with fewer material resources. These findings are consistent with the paradigms of the social determinants of health [15] reinforcing that these elements of marginalization contribute to worse health outcomes requiring more contact with the healthcare system. However, we did see slightly lower rates of in-person care in patients who were newcomers and more racialized. This may be due to factors such as language barriers or a lack of information regarding how to access or navigate appropriate in-person care [16]. Other research reveals there may be fear of deportation leading to avoidance behaviors in immigrants without official status which may lead to choosing virtual forms of care rather than presenting in-person to a healthcare institution [17].

| Score | Quintile | Mean in-person visit rate per 1,000 residents (95% CI) | | Mean virtual visit rate per 1,000 residents (95% CI) | |
|---|---|---|---|---|---|
| ON-Marg Summary Score | 1 | 2,464.11 (2,452.55-2,475.83) | | 2,565.76 (2,549.77-2,582.36) | |
| | 2 | 2,443.98 (2,430.61-2,457.57) | | 2,483.12 (2,462.83-2,502.96) | |
| | 3 | 2,403.84 (2,388.01-2,419.71) | | 2,405.66 (2,386.46-2,424.68) | |
| | 4 | 2,386.32 (2,368.77-2,402.65) | | 2,360.92 (2,342.72-2,380.09) | |
| | 5 | 2,559.87 (2,544.25-2,577.02) | | 2,455.71 (2,434.27-2,476.87) | |
| Age and Labour Force Factor Score | 1 | 2,306.69 (2,293.38-2,319.42) | | 2,523.12 (2,504.82-2,542.18) | |
| | 2 | 2,387.73 (2,375.55-2,401.46) | | 2,471.33 (2,452.43-2,489.08) | |
| | 3 | 2,431.23 (2,416.55-2,445.72) | | 2,420.66 (2,402.52-2,439.04) | |
| | 4 | 2,474.27 (2,458.19-2,488.85) | | 2,383.50 (2,361.66-2,404.04) | |
| | 5 | 2,629.79 (2,612.14-2,650.04) | | 2,443.68 (2,422.41-2,465.42) | |
| Households and Dwellings Factor Score | 1 | 2,471.17 (2,458.29-2,484.71) | | 2,678.63 (2,658.25-2,698.34) | |
| | 2 | 2,443.50 (2,428.87-2,458.59) | | 2,445.20 (2,426.16-2,464.43) | |
| | 3 | 2,420.17 (2,404.23-2,436.23) | | 2,355.90 (2,336.63-2,374.38) | |
| | 4 | 2,445.35 (2,430.57-2,461.66) | | 2,331.48 (2,314.07-2,349.56) | |
| | 5 | 2,456.31 (2,437.33-2,474.56) | | 2,432.74 (2,410.79-2,453.57) | |
| Material Resources Factor Score | 1 | 2,445.47 (2,430.46-2,461.18) | | 2,631.54 (2,614.36-2,651.34) | |
| | 2 | 2,425.89 (2,410.58-2,441.16) | | 2,497.50 (2,477.31-2,517.14) | |
| | 3 | 2,407.53 (2,392.30-2,423.55) | | 2,415.10 (2,396.07-2,435.00) | |
| | 4 | 2,451.34 (2,435.95-2,466.41) | | 2,382.64 (2,363.45-2,402.22) | |
| | 5 | 2,507.10 (2,491.84-2,523.34) | | 2,313.34 (2,292.21-2,331.58) | |
| Racialized and Newcomer Populations Factor Score | 1 | 2,472.78 (2,456.42-2,489.99) | | 2,129.50 (2,111.61-2,148.48) | |
| | 2 | 2,495.72 (2,480.76-2,510.01) | | 2,302.42 (2,284.05-2,319.84) | |
| | 3 | 2,444.56 (2,428.54-2,460.94) | | 2,489.51 (2,469.70-2,508.87) | |
| | 4 | 2,413.67 (2,397.51-2,428.45) | | 2,618.86 (2,601.02-2,638.78) | |
| | 5 | 2,409.36 (2,392.95-2,424.18) | | 2,697.05 (2,676.86-2,715.30) | |

2000 2200 2400 2600 2800    2000 2200 2400 2600 2800

**Fig 1. Comparison of in-person vs virtual care visit rates across the Ontario Marginalization Index individual dimensions and summary score.** (1 = least marginalized, 5 = most marginalized).

Patients who used virtual care during the pandemic were fundamentally different than individuals who were seen in-person. Among most components of the ONMARG index, patients who were the least marginalized had the highest rates of virtual care service use – these were patients with the highest employment-based income, living in stable and cohesive neighborhoods with more financial resources. These realities raise some of the concerns regarding virtual care services in Ontario and equitable accessibility to all patients, particularly among those who are more marginalized.

Interestingly, these trends in rates of virtual care use were reversed for patients from newcomer or racialized populations in whom virtual care use was significantly higher among the most marginalized individuals. The causes for this are not clear and are likely multifactorial. One of the potential explanations for this may include the healthy immigrant effect in which, by virtue of health screening related to Canadian immigration, most newcomers have higher baseline health than the average Canadian population [18]. In keeping with this theory, newcomers to Canada may have had lower acuity presentations leading them to virtual care rather than in-person care [9].

A final explanation which is consistent with the thesis above is that less marginalized patients are more likely to have increased baseline health. As a result, it may have been more appropriate for a higher proportion of these individuals to seek virtual care rather than in-person care. It is not possible to control for patient acuity as, outside of the emergency department, acuity scores are not ascribed to patient visits in the outpatient setting. Future research should seek to establish the drivers of differences in health service utilization and test a variety of interventions to improve access for those who had less access to virtual care.

## Limitations

ONMARG does not address all forms of marginalization that could be experiences or observed within a population [14]. Most notably, ONMARG does not have the ability to isolate the marginalization experienced by Black populations, Indigenous populations, women, and LGBTQ+ communities [14]. Sex-based comparisons could be performed as part of subsequent analyses. The other subgroups discussed above are not well stratified within the provincial administrative databases.

ONMARG uses census data which is only refreshed every 5 years. Within this study, the ONMARG categories referenced for each dimension were dated to 2021. This also means that not everybody is counted as part of the census including the data suppression of People living on reserves, the likely undercount of Indigenous People living off-reserve, the absence of temporary foreign workers and people living in congregate settings (including long-term care).

Lastly, ONMARG is subject to the ecological fallacy. In essence, some individuals may not be represented by their neighborhood averages and are likely misclassified when used as a proxy for their individual socio-economic status. Within this publication, the risk is at its lowest as we used the smallest possible geographic areas (dissemination areas). This said, the risk persists when using these types of datasets.

## Conclusion

Virtual care and in-person care attract different patient populations raising important concerns regarding equitable access. Virtual care seems to attract populations that are more racialized but that are otherwise less marginalized compared to in-person care. These differences much be part of policy considerations for the future of virtual care use and are generalizable outside of the pandemic context.

### Key messages

#### What is already known on this topic –

Access equity in virtual care has been a concern of policy makers, providers, health systems and patients since the shift towards virtual care which was prompted by the COVID-19 pandemic. No quantitative evidence is yet available on access disparities as they relate to patient marginalization.

#### What this study adds –

This study demonstrates that more marginalized populations accessed virtual care significantly less than lesser marginalized populations. This is a trend not observed in analyses of in-person care.

#### How this study might affect policy –

Important differences are seen in virtual care utilization as they relate to patient marginalization which are likely to signify reduced access for more marginalized populations. The contributors to these are multi-factorial and will require a concerted policy effort to measure and overcome.

## Acknowledgments

"This study was supported by ICES, which is funded by an annual grant from the Ontario Ministry of Health (MOH) and the Ministry of Long-Term Care (MLTC). Parts of this material are based on data and/or information compiled and provided by CIHI, the Ontario Ministry of Health, and Ontario Health (OH). This document used data adapted from the Statistics

Canada Postal CodeOM Conversion File, which is based on data licensed from Canada Post Corporation, and/or data adapted from the Ontario Ministry of Health Postal Code Conversion File, which contains data copied under license from ©Canada Post Corporation and Statistics Canada. The analyses, conclusions, opinions and statements expressed herein are solely those of the authors and do not reflect those of the funding or data sources; no endorsement is intended or should be inferred. We thank the Toronto Community Health Profiles Partnership for providing access to the Ontario Marginalization Index. This study was supported by ICES, which is funded by an annual grant from the Ontario Ministry of Health (MOH) and the Ministry of Long-Term Care (MLTC). Parts of this material are based on data and information compiled and provided by CIHI, the Ontario Ministry of Health, and Ontario Health (OH). The analyses, conclusions, opinions and statements expressed herein are solely those of the author. and do not reflect those of the funding or data sources; no endorsement is intended or should be inferred. We thank the Toronto Community Health Profiles Partnership for providing access to the Ontario Marginalization Index.

## Author contributions

**Conceptualization:** Shawn Mondoux, Anastasia Gayowsky, Natasha Clayton, Alim Pardhan, Keerat Grewal.

**Data curation:** Shawn Mondoux, Anastasia Gayowsky.

**Formal analysis:** Shawn Mondoux, Frank Battaglia, Anastasia Gayowsky, Natasha Clayton, Caillin Langmann, Keerat Grewal.

**Funding acquisition:** Shawn Mondoux.

**Investigation:** Shawn Mondoux, Frank Battaglia, Anastasia Gayowsky, Caillin Langmann, Paul Miller, Alim Pardhan, Julie Matthews, Alexander Drossos, Keerat Grewal.

**Methodology:** Shawn Mondoux, Frank Battaglia, Caillin Langmann, Paul Miller, Alim Pardhan, Julie Matthews, Alexander Drossos, Keerat Grewal.

**Project administration:** Shawn Mondoux, Frank Battaglia, Natasha Clayton, Paul Miller, Alim Pardhan, Alexander Drossos, Keerat Grewal.

**Resources:** Shawn Mondoux, Julie Matthews, Keerat Grewal.

**Software:** Anastasia Gayowsky.

**Supervision:** Shawn Mondoux, Paul Miller, Keerat Grewal.

**Validation:** Shawn Mondoux, Natasha Clayton, Keerat Grewal.

**Writing – original draft:** Shawn Mondoux, Frank Battaglia, Natasha Clayton, Caillin Langmann, Paul Miller, Alim Pardhan, Julie Matthews, Alexander Drossos, Keerat Grewal.

**Writing – review & editing:** Shawn Mondoux, Frank Battaglia, Natasha Clayton, Caillin Langmann, Paul Miller, Alim Pardhan, Julie Matthews, Alexander Drossos, Keerat Grewal.

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
