## [Decision Letter · Decision Letter 0]

17 Jan 2025

PONE-D-24-37526A retrospective evaluation of access equity in Virtual Care during the COVID-19 pandemicPLOS ONE

Dear Dr. Mondoux,

Thank you for submitting your manuscript to PLOS ONE. After careful consideration, we feel that it has merit but does not fully meet PLOS ONE’s publication criteria as it currently stands. Therefore, we invite you to submit a revised version of the manuscript that addresses the points raised during the review process.

We look forward to receiving your revised manuscript.

Kind regards,

Pasyodun Koralage Buddhika Mahesh

Academic Editor

PLOS ONE

Journal Requirements:

“Declaration of Interest: SM was the recipient of the Juravinski Research Institute grant which paid for the costs of administrative data use. SM was also the recipient of the Graham Farquharson Knowledge Translational Fellowship, administered by PSI, which created protected time for research. NC was a paid research coordinator as part of this project. SM, FB, CL, PM, AP, AD and KG are all clinicians providing clinical care using both virtual and in-person modalities throughout the pandemic. AG is a paid employee of the Institute of Clinical Evaluative Sciences (ICES). “

4. Please expand the acronym “PSI” (as indicated in your financial disclosure) so that it states the name of your funders in full.

5. Please upload a copy of Figure 1, to which you refer in your text on page 5. If the figure is no longer to be included as part of the submission please remove all reference to it within the text.

6. Please include a copy of Tables 1 and 2 which you refer to in your text on page 7.

Reviewers' comments:

Reviewer's Responses to Questions

**Comments to the Author**

1. Is the manuscript technically sound, and do the data support the conclusions?

Reviewer #1: Yes

Reviewer #2: Yes

2. Has the statistical analysis been performed appropriately and rigorously? 

Reviewer #1: Yes

Reviewer #2: No

3. Have the authors made all data underlying the findings in their manuscript fully available?

Reviewer #1: No

Reviewer #2: Yes

4. Is the manuscript presented in an intelligible fashion and written in standard English?

Reviewer #1: Yes

Reviewer #2: Yes

5. Review Comments to the Author

Reviewer #1: The manuscript discusses an important timely topic. However, there are some areas that you need to pay attention to improve the clarity of the manuscript.

Title: A retrospective evaluation of access equity in virtual care during the COVID-19 pandemic (EVOQUe - equity)

The manuscript addresses an important and timely topic.

Suggestions for improvement: The study sample has to be mentioned in the title. Do not use statements within brackets in the title.

Introduction: The introduction provides a clear background on the growing use of virtual care in Canada, emphasizing its accelerated adoption during the pandemic and the challenges of equitable access. The authors have identified the gap in the availability of quantitative data addressing virtual care disparities.

However, the manuscript will further benefit from a clear definition for the word ‘Virtual care’. What type of virtual care were you referring to? Is this research question only relevant to the health programs carried out in virtual platforms or any other?

The background of the available literature on and around the topic should be elaborated. What made you think that virtual platforms do not support equitable access?

Please state the research question explicitly. What specific aspects of equity have you prioritized in this study? (e.g. socio-economic, geographic, ethnic etc.) What do you mean by marginalized population?

Are the results generated from this study only relevant to the pandemic times or address broader implications for future virtual care policy?

Methods

Why were some of these groups excluded from the study? Won’t that create any bias to the final result?

Discussion

Provides a good interpretation of the study findings. This part of the manuscript can benefit from critically analysing the present findings with the findings from previous literature from both international and Canadian setting.

Reviewer #2: Introduction

The authors have provided a clear rationale for conducting the current study, in a context where there is limited availability of studies assessing the equity in virtual care access quantitatively, despite the availability of qualitative data. It is suggested to also provide a brief justification for selecting the ONMARG scores for the purposes of the current study.

The aim of the study has been given as ‘to explore factors influencing the accessibility virtual care and disparities faced by various population groups’ in the introduction. This is somewhat in contrast to what is mentioned in the abstract, which mentions, ‘a comparison of healthcare utilization between virtual care and in-person care in the province of Ontario across all dimensions of the Ontario Marginalization Index and compared the rates of use within each’, which is evident as the methodology that has been followed. This difference in the depiction of the objective of the study might create some confusion to the reader initially in the text. Hence, it is suggested to mention the aim/ objective of the study in a consistent way throughout the manuscript.

Methods

Setting and population

The setting and the timeframe of analysis have been clearly defined. Exclusion criteria have been defined in the manuscript. It has been mentioned that patients without a valid Ontario health card have been excluded from the analysis. It is worth mentioning the rationale for this exclusion, how much of a proportion this category was out of the total visits, and whether excluding them would have created any bias in the study.

It is stated that there was a very high proportion of ‘undefined’ location codes within the ICES billing data. It is not very clear whether these visits were excluded or whether they were included following review. Suggest elaborating it to give a clear idea.

Suggest including a brief rationale for conducting the analysis on a per-visit basis rather than on a per-patient basis, for the clear understanding of the reader.

Data sources

The different databases linked for extracting the data required for the analysis have been well-explained.

Outcome measures and analyses

The outcome measure has been given as the mean visit rate per 1000 residents within the dissemination area. It would be clearer to the reader if it was specified as the ‘in-person or virtual’ visit rate. Further, it is not very clear whether each dissemination area has been taken as the unit measure in the analysis. Hence, it is suggested to clearly mention it. Further, It would be good to mention the number of dissemination areas included in the study altogether. Furthermore, the analysis method that has been carried out in comparing the mean visit rate (either in-person or virtual) in the dissemination areas belonging to different marginalization categories is not clear. It is suggested to explain this further for the benefit of the reader.

Ethics and approval

The rationale for not requiring informed consent and ethics approval have been explained.

Results

Study population

Table 1 and 2 were not available in the manuscript, for review. Hence, it is not possible to comment on this section. Suggest including the tables.

Access equity

An elaborated explanation has been provided on the comparison of utilization of virtual care (and in-person care) across different levels of marginalisation. However, it is not clear if these interpretations of the results are based on the comparison of descriptive mean values obtained as mean visit rates, or whether further analytical methods have been used for the comparison. Suggest presenting the available data values of the data analysis that has been carried out. Figure 1 was not available in the manuscript, for review, hence it is not possible to comment on the results/ data analysis. Suggest including Figure 1.

Interpretation

The results obtained in the study have been interpreted considerably in terms of the differences observed in virtual and in-person visits at different levels of marginalisation. However, in my view, saying that ‘Patients who used virtual care during the pandemic were fundamentally different than individuals who were seen in-person’ would require a more robust analysis of data to justify it.

The explanation given paragraphs 3 and 4 mention about the healthy immigrant effect and the likelihood of increased baseline health of less marginalized patients, as one of the causes/ explanations for seeking virtual care in contrast to in-person care. In my view, it would be good to mention the other possible causes suggested for the differences as well, to provide the reader a more comprehensive interpretation.

Limitations

Limitations of the study have been well-identified and explained.

Conclusion

It would be good to provide some recommendations for future research and elaborate some more on the policy considerations mentioned.

6. PLOS authors have the option to publish the peer review history of their article (what does this mean? ). If published, this will include your full peer review and any attached files.

**Do you want your identity to be public for this peer review?** For information about this choice, including consent withdrawal, please see our Privacy Policy .

Reviewer #1: No

Reviewer #2: No

---

## [Author Response · Author response to Decision Letter 1]

18 Feb 2025

These were included in tabular format within the submitted document. Many thanks!

---

## [Decision Letter · Decision Letter 1]

6 Apr 2025

A retrospective evaluation of access equity in Virtual Care during the COVID-19 pandemic

PONE-D-24-37526R1

Dear Dr. Mondoux,

We’re pleased to inform you that your manuscript has been judged scientifically suitable for publication and will be formally accepted for publication once it meets all outstanding technical requirements.

Kind regards,

Pasyodun Koralage Buddhika Mahesh

Academic Editor

PLOS ONE

Additional Editor Comments (optional):

Reviewers' comments:

Reviewer's Responses to Questions

**Comments to the Author**

1. If the authors have adequately addressed your comments raised in a previous round of review and you feel that this manuscript is now acceptable for publication, you may indicate that here to bypass the “Comments to the Author” section, enter your conflict of interest statement in the “Confidential to Editor” section, and submit your "Accept" recommendation.

Reviewer #1: All comments have been addressed

Reviewer #2: All comments have been addressed

2. Is the manuscript technically sound, and do the data support the conclusions?

Reviewer #1: Yes

Reviewer #2: Yes

3. Has the statistical analysis been performed appropriately and rigorously? 

Reviewer #1: Yes

Reviewer #2: Yes

4. Have the authors made all data underlying the findings in their manuscript fully available?

Reviewer #1: Yes

Reviewer #2: Yes

5. Is the manuscript presented in an intelligible fashion and written in standard English?

Reviewer #1: Yes

Reviewer #2: Yes

6. Review Comments to the Author

Reviewer #1: The study is conducted on an interesting topic and has followed a technically sound methodology. Authors have addressed all the comments made during the previous review. The manuscript is now suitable for publication.

Reviewer #2: The comments given for the previous version of the manuscript have been satisfactorily addressed by the authors.

7. PLOS authors have the option to publish the peer review history of their article (what does this mean? ). If published, this will include your full peer review and any attached files.

**Do you want your identity to be public for this peer review?** For information about this choice, including consent withdrawal, please see our Privacy Policy .

Reviewer #1: No

Reviewer #2: No

---

## [Editor Report · Acceptance letter]

PONE-D-24-37526R1

PLOS ONE

Dear Dr. Mondoux,

I'm pleased to inform you that your manuscript has been deemed suitable for publication in PLOS ONE. Congratulations! Your manuscript is now being handed over to our production team.

Kind regards,

on behalf of

Dr. Pasyodun Koralage Buddhika Mahesh

Academic Editor

PLOS ONE